# In Vitro Bioaccessibility of Selenium from Commonly Consumed Fish in Thailand

**DOI:** 10.3390/foods11213312

**Published:** 2022-10-22

**Authors:** Alongkote Singhato, Kunchit Judprasong, Piyanut Sridonpai, Nunnapus Laitip, Nattikarn Ornthai, Charun Yafa, Chanika Chimkerd

**Affiliations:** 1Doctor of Philosophy Program in Nutrition, Faculty of Medicine Ramathibodi Hospital and Institute of Nutrition, Mahidol University, Nakhon Pathom 73170, Thailand; 2Institute of Nutrition, Mahidol University, Salaya, Phutthamonthon, Nakhon Pathom 73170, Thailand; 3Chemical Metrology and Biometry Department, National Institute of Metrology (Thailand), Pathum Thani 12120, Thailand

**Keywords:** selenium, in vitro, bioaccessibility, fish, equilibrium dialyzability

## Abstract

Selenium (Se), abundantly obtained in fish, is a crucial trace element for human health. Since there are no data on Se bioaccessibility from commonly consumed fish in Thailand, this study assessed the in vitro bioaccessibility of Se using the equilibrium dialyzability method. The five fish species most commonly consumed in Thailand were selected to determine total Se content using several preparation methods (fresh, boiling, and frying). Equilibrium dialyzability was used to perform in vitro bioaccessibility using enzymatic treatment to simulate gastrointestinal digestion for all boiled and fried fish as well as measuring Se using inductively coupled plasma triple quadrupole mass spectrometry (ICP-QQQ-MS). Two-way ANOVA with interaction followed by Tukey’s honestly significant difference (HSD) post hoc test revealed that boiled Indo-Pacific Spanish mackerel, longtail tuna, and short-bodied mackerel were significantly higher in Se content than striped snakehead and giant sea perch (*p* < 0.05). For fried fish, longtail tuna showed the highest Se content (262.4 µg/100 g of product) and was significantly different compared to the other fish (*p* < 0.05, estimated marginal means was 43.8–115.6 µg/100 g of product). Se bioaccessibilities from striped snakehead (70.0%) and Indo-Pacific Spanish mackerel (64.6%) were significantly higher than for longtail tuna (*p* < 0.05). No significant difference in bioaccessibility was found in terms of preparation method (i.e., boiling and frying). In conclusion, the fish included in this study, either boiled or fried, have high Se content and are good sources of Se due to high bioaccessibility.

## 1. Introduction

Selenium (Se) is a crucial nutrient for human health. Adequate Se intake supports physiological functions and lowers the risk of complications by downregulating T cell differentiation and reactive oxygen species (ROS) [1,2]. The cut-off criteria for Se deficiency is serum Se below 63 µg/L [3]. Thailand’s Dietary Reference Intake (DRI) guideline recommends Se intake of 55 µg daily for adults [4]. Unfortunately, many countries still report Se deficiency. For instance, Spanish children (13.9%) were reported as having a low level of serum Se (<60 µg/L) [5], adults in Saudi Arabia (41%) had low Se concentration (<0.56 µg/L) in toenails [6], and children living with human immunodeficiency virus (HIV) in Thailand (56%) were reported as having low serum Se [7]. Long-term inadequate Se intake is related to impaired immunological functions and the risk of carcinogenesis [8]. Hence, consuming foods that are high in Se should be promoted to prevent Se insufficiency among populations.

Widely consumed around the world, fish are well-known as good sources of Se, and previous studies have reported high Se contents in several species of fish. Most notably, in Canada, Se concentrations in Wahoo and flying fish were 1.4 and 0.8 mg/kg, respectively [9]. A Japanese study found 0.75, 0.77, and 1.27 mg/kg of Se in Kuromaguro, Musu, and Kinmedai fish, respectively [10]. Furthermore, Se concentrations in Atlantic herring, Atlantic halibut, and Atlantic cod were 0.38–0.61, 0.47–0.48, and 0.25–0.31 mg/kg, respectively, according to a European study [11]. Thailand is a large fish-producing nation due to extensive offshore fishing and fish farming. Due to inexpensive prices, fish are easily accessible and affordable sources of protein for the Thai people and are included in a variety of Thai traditional menus. A previous study reported high Se content in fish that are commonly consumed in Thailand, such as longtail tuna, short-bodied mackerel, Indo-Pacific Spanish mackerel, and giant sea perch at 208.4, 80.7, 89.7, and 47.9 μg/100 g edible part, respectively [12].

Bioaccessibility is defined as the content of nutrients released from a food matrix after ingestion in the human gastrointestinal system that are then available for further absorption through the gut barrier. Hence, foods with high bioaccessibility of specific nutrients can have potentially high bioavailability and thus can enter the circulatory system and reach target physiological functions [13]. Consequently, an enhanced understanding of the bioaccessibility of nutrients in foods can help people in general, and especially those groups at risk of Se deficiency, to improve their food intake and optimize the efficiency of nutrients for their bodily functions. 

The bioaccessibility of nutrients can be determined by the in vitro method that mimics the human digestive system to provide information on the interaction of nutrients, the food matrix, methods of food processing and preservation, luminal factors including enzymes and pH, food components, etc. Moreover, this method is useful for evaluating potential micronutrient absorbability [14]. The in vitro method has been used to determine nutrient bioaccessibility in previous studies, such as levels of calcium in vegetables low in phytate, dietary fiber, and oxalate including kale, collard, and soybean sprouts (20–39%) [15]. Another study investigated the in vitro bioaccessibility of iron in colored and white beans, i.e., 1.5–2.7% and 12.1–18.8%, respectively [16]. For Se in fish, a previous European study reported high in vitro bioaccessibility in tuna, swordfish, and sardines at 50, 76, and 83%, respectively [17]. While a previous study established the Se content in fish commonly consumed in Thailand, to our knowledge no data exist on the bioaccessibility of Se from those fish [12]. Consequently, this study investigates the bioaccessibility of Se based on the in vitro method using the equilibrium dialysis technique from fish widely consumed among the Thai people and using inductively coupled plasma triple quadruple mass spectrometry (ICP-QQQ-MS), the preferred analytical technique due to its lower detection limit values and minimization of interferences.

## 2. Materials and Methods

### 2.1. Chemicals and Reagents

Sigma-Aldrich provided high purity grade nitric acid (Suprapure 65% HNO_3_), calcium disodium ethylene-diaminetetraacetate hydrate (EDTA-Ca-Na_2_), pepsin from porcine stomach mucosa (P-7000), pancreatin from porcine pancreas (P-1750), and bile extract porcine (B-6831) (St Louis, MO, USA). The National Institute of Standards and Technology (NIST), Gaithersburg, MD, USA, provided stock solutions of Se (SRM3149), CRM SRM1566b (oyster tissue), and Rh (Rh SRM3144). In addition, this study made use of CRM from the National Metrology Institute of Japan (NMIJ7402-a, codfish tissue). Merck provided potassium hydroxide (KOH, 105033), sodium bicarbonate (NaHCO_3_), and dialysis membrane (molecular weight cut-off 12–14 KDa, 13 mm internal diameter) (KGaA, Darmstadt, Germany). Medicell Membranes Ltd. provided flat dialysis membranes (MWCO 12–14 kDa, 15.9 mm wide) (Greenwich, London). In the study, deionized water from Milli-Q^®^ water (Millipore Sigma, Burlington, MA, USA) was employed.

### 2.2. Sample Preparation

The five most commonly consumed fish in Thailand (both freshwater and marine fish) according to survey results [18] and the Thai food composition database [19] were purchased from three randomly selected markets located in and nearby Bangkok (Klong Toey market, Tai market, and Bangkok Noi market). Each fish species at each market was randomly selected from 3–4 vendors to be representative of each sample. Table 1 presents the common names of the fish sampled in this study, as well scientific names and other information. All fish samples were stored in cooling bags to retain freshness after purchase and until being sent to the Institute of Nutrition, Mahidol University, to be prepared as boiled and fried fish and then homogenized (with skin), after which the moisture content was evaluated according to procedures described elsewhere [20,21]. Briefly, the weights of all fish before and after eliminating inedible parts were recorded. The fish cooked by boiling were boiled in deionized water, while the fish prepared by frying were fried in palm oil. Portions of all samples were dried using a freeze-dried system (freeze dryer model FD-5, Genvac, Bangkok, Thailand) and ground using a grinder (Philips, 600 W, Indonesia) into fine particles. The fish were then kept in aluminum foil bags and stored at −20 °C.

### 2.3. Weight Yield Factor

The weight yield factor is the change in weight caused by the loss or increase of water and/or fat in fish after cooking. It is determined by dividing the weight (g) of cooked fish by the weight (g) of raw fish.

### 2.4. Determination of Se Concentration

At the National Institute of Metrology in Thailand, all fish species were analyzed for total Se concentrations in fresh, fried, and boiled forms using inductively coupled plasma triple quadrupole mass spectrometry (ICP-QQQ-MS) (Agilent 8800 triple quadrupole ICP-MS, Agilent Technologies, Santa Clara, CA, USA). Collision and reaction gas modes were used to decrease interferences during ICP-QQQ-MS analyses and to improve selenium sensitivity and accuracy. An external calibration curve was created by a series of working standard solutions of Se in 2% *v*/*v* HNO_3_. An internal standard (Rh) was also prepared with 2% *v*/*v* HNO_3_, which was pipetted into digested sample solutions at the same concentration to monitor any change in instrument condition during analysis.

The procedures for microwave digestion and final solution preparation of the fish samples were adapted from a previous study [22]. Briefly, all freeze-dried fish samples (0.5 g) and CRMs (0.25 g) were weighed and added to glass vials followed by 5 mL of concentrate HNO_3_ (65% *v*/*v*). The mixed samples then underwent microwave-assisted digestion in triplicate using Anton Paar Multiwave 7000 (Anton Paar, Graz, Austria) with conditions shown in Table 2. Thereafter, each digested sample was made to volume with DI water in another volumetric vial (40 mL). These volume aliquots were then pipetted into other volumetric vials to which an internal standard (Rh) at the same concentration (final volume was 10 mL) was added before being injected into the ICP-QQQ-MS. To avoid any sample contamination during preparation, blank samples were also prepared using the same procedures as the fish samples (digested only 65% HNO_3_).

### 2.5. True Retention of Se in Cooked Fish

True retention (TR) is the proportion of a nutrient that remains in a cooked food as compared to the original quantity prior to cooking. As a result, TR indicates the influence of different cooking methods on the amount of nutrient change after processing, which is reported as the percentage of change. Each fish species was weighed (3 significant digits) and recorded before and after cooking for this experiment: %TR = (µg Se of cooked fish × weight of cooked fish/µg Se of raw fish × weight of raw fish) × 100.

### 2.6. Determination of In Vitro Bioaccessibility of Se

Traditional Thai fish menus/recipes usually include cooked fish; consumption of the selected fish species in raw form is very rare. Consequently, the gastrointestinal simulation procedure was performed only on boiled or fried fish using the equilibrium dialysis system according to a previous study [23]. In brief, 0.5 g of each cooked fish sample was weighed in triplicate and placed in an Erlenmeyer flask, followed by 95 mL of deionized water, and the pH of 2.0 adjusted with HCl. Deionized water was used to adjust the final volume to 100 mL. Thereafter, each flask received 3 mL of fresh pepsin solution made by dissolving 16 g of pepsin in 100 mL of 0.1 M HCl. Flasks were incubated in a shaking water bath (Memmert GmbH, Germany) (37 °C for 2 h). Titratable acidity was determined as the volume of KOH needed to titrate a pepsin digest and pancreatin bile extract mixture to pH 7.5. The digesta’s titratable acidity was measured by pipetting each aliquot sample of 20 mL into another flask, followed by 5 mL of fresh pancreatin-bile extract solution (dissolving 0.4 g of pancreatin and 2.5 g of bile extract in 100 mL of 0.1 M NaHCO_3_). The solution was then titrated with 0.5 N KOH to pH 7.5, and the quantities of KOH needed to achieve each sample’s target pH were recorded. Once incubation was completed, the incubated samples were pipetted at 20 mL to other 50 mL volumetric vials in triplicate. For this study, dialysis bags were prepared by rinsing several times and soaking in deionized water for 10 min, boiled with 40% ethanol for 10 min, rinsed with deionized water mixed with 0.01 M EDTA-Ca_2_ and 2% NaHCO_3_, rinsed again with deionized water, then soaked in 0.001 M NaHCO3 and stored at 4 °C until use. For the analysis, the dialysis bags contained identical volumes of 0.5 N NaHCO_3_ equivalent to 0.5 N KOH used to analyze for titratable acidity in each fish sample. A final volume of 7 mL was made with deionized water and placed inside 50 mL volumetric vials that contained an aliquot (20 mL) of pepsin digesta. The vials were then incubated for 30 min in a shaking water bath at 37 °C. Thereafter, a prepared pancreatin bile extract mixture of 5 mL was added into the incubating vials, and incubation was continued for 2 h. Reagent blanks were also conducted to control for checking and corrected contamination.

In the end, the dialysates and residue digesta of the fish samples were weighed and assessed for Se content using ICP-QQQ-MS. Briefly, the dialysates and digesta were dried to 1 mL and then digested using microwave digestion as per the described procedure and conditions shown in Table 2. Thereafter, the digested dialysates and digesta were dried again until reaching 1 mL and made to a volume of 5 mL with deionized water, and then filtered using a 0.22 µm nylon filter with a syringe (Merck Millipore) before analysis. Dialyzability was presented as the percentage of Se content in the dialyzate to the total amount of Se obtained in the sample as follows: dialyzability (%) = 100 × dialyzed Se content (µg/g sample) divided by the total Se content (µg/g sample). Digested residue digesta were also assessed for Se content using ICP-QQQ-MS to achieve the mass balance analysis.

### 2.7. Analytical Method Precision and Accuracy

To confirm and ensure the precision of the Se analysis using ICP-QQQ-MS, CRMs from NMIJ7402-a (codfish tissue) and SRM1566b (oyster tissue) were digested and analyzed in triplicate. CRM Se concentrations in dry matter were determined in accordance with each CRM’s certified value. These CRMs were used to evaluate the accuracy and reliability of Se determination measurements. The experimental Se concentration for NMIJ7402-a in this study was 1.93 0.07 mg/kg (certified value at 1.80 0.20 mg/kg) and for SRM1566b was 2.08 0.13 mg/kg (certified value at 2.06 0.15 mg/kg) [21]. There were no significant differences between analyzed values and certified values of the CRM (*p* > 0.05). The percentage of relative standard deviation (%RSD) was assessed to determine the method precision. In this study, %RSD of CRMs was 4.00% for NMIJ7402-a and 6.59% for SRM1566b [21], which indicated good precision. Consequently, the method for Se analysis provided accurate and precise results.

### 2.8. Limit of Detection and Limit of Quantification in This Study

The estimated LOD (lowest concentration of Se in fish samples) and LOQ (lowest quantification of Se in fish sample) using the ICP-QQQ-MS method were previously studied and reported elsewhere [21]. The method LOQ was 3 µg/kg with obtained good precision (6.2% RSD), which passed the acceptable LOQ criteria [24,25]. 

### 2.9. Statistical Analysis

Mean standard deviation (SD) of yield factors, percentages of edible portions, moisture contents, true retention, and percentage of dialyzability of boiled and fried fish from three sample sources for Se concentrations and Se content in g/100 g of product were measured. At *p* < 0.05, two-way ANOVA followed by Tukey’s Honestly Significant Difference was used to establish the statistical significance of Se contents for the different types of fish, their true retention after cooking, and percentages of dialyzability. For statistical analysis, IBM^®^ SPSS Statistics for Windows, Version 21.0, was applied.

## 3. Results

### 3.1. Total Se Concentrations of Fish

Total Se concentrations of the selected fish prepared using common cooking methods are shown in Table 3 along with data on edible portion, yield factor, moisture content, and true retention. In general, fried longtail tuna had the highest Se content (262.4 µg/100 g product). No significant difference was found in terms of total Se concentration in the same fish species using different cooking methods. High true retention of Se among selected fish (61.4–100%TR) reflects loss of Se after boiling and frying. The edible portion was between 41 and 73% with a yield factor of 0.7–0.9.

Longtail tuna had the highest Se content for any cooking method. The combined effects, or the changes that one thing produces in another, of fish species and cooking practices on Se content as evaluated by two-way ANOVA with interactions followed by Tukey’s HSD post hoc test were statistically different (*p* < 0.05) (Figure 1A). Furthermore, data for boiled fish showed that longtail tuna, Indo-Pacific Spanish mackerel, and short-bodied mackerel contained considerably more Se than striped snakehead and giant sea perch (*p* < 0.05). Longtail tuna had the highest Se concentration among the fried fish (262.4 µg/100 g of product) and was significantly different from the other fish (*p* < 0.05, estimated marginal means were 43.8–115.6 µg/100 g of product) (Table 4).

The effect of cooking methods on Se content was also significantly different (*p* < 0.05) between fried fish (estimated marginal means was 110.2 µg/100 g of product) and boiled fish (estimated marginal means was 81.2 µg/100 g of product) (*p* < 0.05) (Table 5).

### 3.2. The Effect of Different Cooking Methods on Fish %TR

The data on %TR represented the percentage of Se that remained in cooked fish as compared to the Se content before cooking. Two-way ANOVA with interaction, followed by Tukey’s HSD post hoc test, demonstrated that the combined effects of fish species and cooking procedures on %TR were significantly different (*p* < 0.05) (Figure 1B). For instance, the %TR of boiled longtail tuna had the lowest value (64.4%). Longtail tuna and striped snakehead were also found to be significantly lower (*p* < 0.05) than giant sea perch (100%TR), short-bodied mackerel (88.8%TR), and Indo-Pacific Spanish mackerel (78.8%TR). Short-bodied mackerel had the lowest %TR (64.1%) for the frying method, whereas Indo-Pacific Spanish mackerel and longtail tuna had 100%TR. The two-way ANOVA with Tukey’s HSD post hoc test revealed that Indo-Pacific Spanish mackerel, longtail tuna, and striped snakehead had significantly greater (*p* < 0.05) %TR than giant sea perch and short-bodied mackerel (Table 4). The percentage of TR difference between boiling and frying was likewise considerable (Table 5). Fried fish had a considerably higher %TR (*p* < 0.05, calculated marginal means were 88.5%TR) than boiled fish (83.4%TR). 

The percentages of Se bioaccessibility using the in vitro method for the selected fish are presented in Figure 2. Results ranged between 49.0 and 70.0% for boiled fish and between 47.1 and 58.5% for fried fish. Hence, fish prepared by frying were slightly lower than those prepared by boiling in terms of Se bioaccessibility using the in vitro method analyzed by ICP-QQQ-MS. However, no significant difference was found in terms of percentage bioaccessibility in the same fish species prepared using different cooking methods (boiling and frying). Boiled striped snakehead had the highest percentage of Se bioaccessibility compared to other fish. According to two-way ANOVA with interactions followed by Tukey’s HSD post hoc test (*p* < 0.05), Se bioaccessibility levels in boiled striped snakehead (70.0%) and boiled Indo-Pacific Spanish mackerel (64.6%) were significantly greater than those of boiled longtail tuna (49.0%). No significant difference was found in the frying method on Se bioaccessibility among the fish samples. Moreover, the cooking method had no effect on the overall percentage of Se bioaccessibility, and there was no significant difference across boiling and frying (Table 6). The accuracy of precision on the equilibrium of Se content in dialyzability and residue was ensured by performing mass balance analysis. The percentage of recovery Se in boiled fish was 91.7 ± 0.4, and that in fried fish was 104.8 ± 7.8 compared to the analyzed total Se content (data not shown in the Table).

## 4. Discussion

### 4.1. Edible Portion (EP) and Total Se Concentrations of Fish 

The EP range is dependent on the actual preparation method used for the fish. In Thai household practice, the internal organs and scales of fish are removed before cooking, which may explain why the EP of the selected fish in this study was different from results previously presented for Bangladesh (62–85%) [26]. Moreover, the yield factor is influenced by several fish tissue components, which may result in weight changes after cooking [27]. In addition, different fat contents obtained in foods could be associated with yield factor [28]. The lowest moisture content, according to this analysis, was found in fried giant sea perch (54 g/100 g) while the highest was in short-bodied mackerel (76 g/100 g). This finding is in line with the loss of moisture that occurs more often in fried fish compared to boiled fish due to heat processing at high temperatures, which impacts heavily on the loss of moisture in foods, especially for fish [29,30,31,32].

The results for total Se concentration support the findings of a previous study that indicated frying and heat processing at high temperatures had no effects on the degradation of Se in Se-biofortified grains when compared to traditional Chinese cooking methods [33]. The general findings of this present study are comparable to those for other areas, such as fish in Japan, where Se concentrations ranged from 12 to 127 µg/100 g [10], and in European fish, where they ranged from 22 to 61 µg/100 g [11]. In addition, the findings reveal that Se is different from other nutrients obtained in fish that are commonly consumed in Thailand, such as vitamin D content, which is not affected by the cooking method [21]. Alternatively, the diverse Se forms found in fish may possibly be the primary difference influencing Se loss after cooking. According to prior research, the primary forms of Se obtained in most fish are selenocysteine and selenomethionine [34]. Selenomethionine is the most common chemical type of Se seen in tuna blood [35]. These Se forms have a low molecular weight and dissolve quickly in water upon boiling [36], which could explain why boiling causes more Se loss than frying.

### 4.2. The Impact of Various Cooking Processes on Fish%TR

Most of the sampled fish in this study were marine fish with no scales. As a result, in comparing fish varieties, caution must be taken because the presence of scales may reduce heat exposure during cooking [37,38]. In general, this study’s findings agree with information from the Food and Agriculture Organization of the United Nations (FAO) on the genuine preservation of Se in fish after processing, where a range of 90–100% demonstrated that Se is resistant to heat [39]. In addition, the results of %TR are comparable with a study in Europe that indicated steamed Gilthead seabream (*Sparus aurata*) obtained a %TR of Se of 90–100% [40].

### 4.3. In Vitro Bioaccessibility of Se in Fish

This present study’s results on the high bioaccessibility of Se among fish species commonly consumed in Thailand are comparable to those of European fish, such as sardine (83%), swordfish (76%), and tuna (50%) [17]. These findings indicate that the selected fish species used in this study have potentially high efficiency in terms of Se absorption and Se bioavailability in the human body, since the Se obtained from these fish are highly released from matrix tissue by human gastrointestinal simulation [41]. Consequently, nutritionists should recommend these fish for healthy people, in general, and especially for groups at risk of Se deficiency. In addition, a previous study reported that macro-elements obtained from fish tissues play an important role in the bioaccessibility of Se. Se bioaccessibility is positively related to carbohydrate content and dietary fiber [42]. In this study, the fish contained a low amount of carbohydrate (0–1 g/100 g, data not presented) and no dietary fiber. Protein content exhibits a negative correlation with Se bioaccessibility but no correlation between the bioaccessibility fraction of Se and the fat content [42]. In this study, all of the fish had protein contents of 22.6–26.7 g/100 g for boiling and 29.6–24.9 g/100 g for frying (detailed data not presented). They also provided fat contents of 2.1–5.6 g/100 g for boiling and 6.9–13.8 g/100 g for frying. A negative correlation was found between Se bioaccessibility and protein (r = −0.65) and fat (r = −0.41). In addition, a previous study reported that selenomethionine is the main chemical form of Se in fish, and it is the major form of Se in fish dialysate [43]. However, there is no information on selenium form in this study as yet.

Potentially toxic element contamination in fish tissues and other seafood products is common [44], especially mercury (Hg), which is most commonly in the form of methylmercury (MeHg). Se is recognized as decreasing the toxicity of MeHg. The present data established that the Se:Hg molar ratio in the fish was 0.23:1 [45]. In addition, a previous study revealed that a high ratio of Se/Hg in fish was related to low bioaccessibility of MeHg according to the in vitro method [46]. Other potentially toxic elements reported as usually contaminating fish and other seafood products are cadmium, arsenic, and lead, which are harmful to humans [47]. Long-term consumption of foods that are contaminated with these elements can increase the risk of cancer development [48]. Consequently, determination of potentially toxic element contents and their bioaccessibility in fish commonly consumed in Thailand is an area for future study to ensure that the fish are safe to consume with below the maximum levels of potentially toxic element concentrations (i.e., Pb < 0.3 mg/kg, Hg < 1.2–1.6 mg/kg, etc.) based on the criteria established by the FAO [49].

Finally, Se can come in a variety of chemical forms. Organic forms, most commonly found as selenomethionine and selenocysteine, are prevalent in fish and other animal products [50,51]. Inorganic forms, mainly selenite and selenate, have been reported to be poisonous and ineffective in the human body [52,53]. Since in our study a Se speciation analysis in fish dialysate was not undertaken, a future study focusing on this analysis is needed to determine which major Se form was obtained in the dialysate, and to determine its potential to be absorbed from these selected fish samples.

## 5. Conclusions

This study’s results indicate that the selected fish are excellent sources of Se that can be consumed and promoted for consumption in either boiled or fried forms, as well as in raw form. Although there is some loss of selenium after cooking, nonetheless, the cooked fish provided high levels of Se. This study is the first to report selenium bioaccessibility in Thailand due to the difficulty of selenium analysis. Se bioaccessibility for all of the studied fish is about one-half to two-thirds of the original Se content. However, these fish also showed a high Se content after cooking as well as with in vitro bioaccessibility. Consequently, an in vivo human study of Se bioavailability should be further conducted. In conclusion, all five studied fish (striped snakehead, giant sea perch, Indo-Pacific Spanish mackerel, longtail tuna, and short-bodied mackerel) are highly recommended for consumption, especially among those persons who might be Se deficient.

## Figures and Tables

**Figure 1 foods-11-03312-f001:**
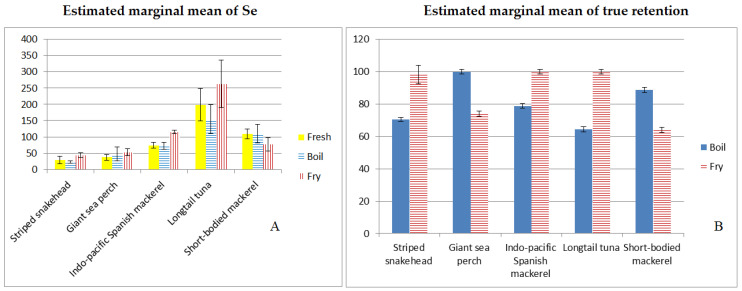
The combined effects of several fish species and cooking methods on Se concentration; y-axis is µg/100 g of product (**A**) and percentage of true retention; y-axis is means of percentage (**B**).

**Figure 2 foods-11-03312-f002:**
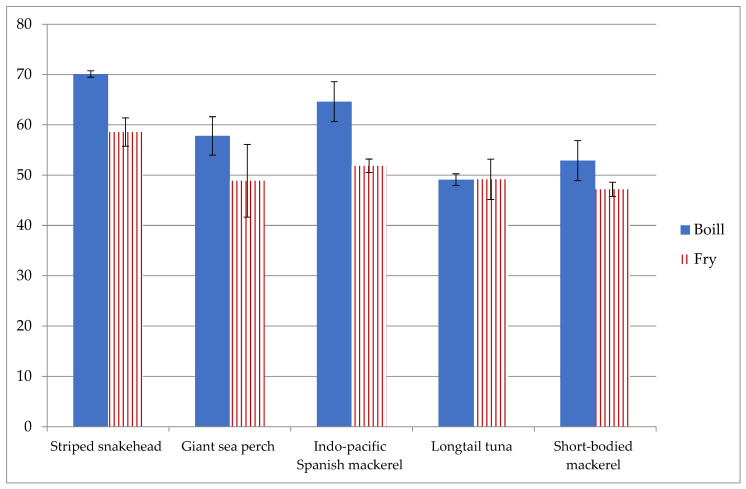
Value of percentage of in vitro bioaccessibility of Se obtained in boiled and fried fish.

**Table 1 foods-11-03312-t001:** The selected top 5 most commonly consumed fish used in this study.

Common Name	Fish with Scale	Scientific Name	Local Name	Habitat	Purchase
					(Month/Year)
Striped snakehead	Yes	*Channa striata*	Pla-chon	Freshwater	August 2018
Giant sea perch	Yes	*Lates calcarifer*	Pla-kha-pong-khaw	Marine	September 2018
Indo-Pacific Spanish mackerel	Yes	*Scomberomorus guttatus*	Pla-in-see	Marine	September 2018
Long tail tuna	No	*Thunnus tonggol*	Pla-O	Marine	December 2018
Short-bodied mackerel	No	*Rastrelliger brachysoma*	Pla-tu	Marine	September 2018

**Table 2 foods-11-03312-t002:** Parameters used for digestion and analysis of total selenium concentration.

Methods	Setting	
**Microwave system parameter:**		
Estimated sample weight	0.5 g	
Starting pressure	40 bar	
Pressure	160 bar	
Step time	Step 1: 25–220 °C, 20 min Step 2: 220 °C, 20 min	
Cooling temperature	50 °C	
Pressure release rate	8.0 bar/min	
**ICP-QQQ-MS parameter:**	He (99.999%) mode	O_2_ (99.999%) mode
Radio frequency (RF) power	1550 W	
Sampling depth	8 mm	
Carrier gas, flow rate	1.05 L/min	
Makeup gas, flow rate	0.2 L/min	
	3 mL/min	30%
Monitor masses	^77^Se, ^78^Se, ^82^Se, ^78^Se^16^O^+^, ^80^Se^16^O^+^, ^82^Se^16^O^+^	

**Table 3 foods-11-03312-t003:** Data on percentage of edible portion, yield factor, and moisture content of 3 individual samples (expressed as mean ± SD) from each type of selected fish.

Fish Name	Sample	Edible Portion (%)	Yield Factor	Moisture (g/100 g)	Se Concentration (µg/100 g of Product)	True retention of Se (%)
Striped snakehead	Fresh (with skin)	50 ± 3	-	74 ± 0.4	29.4 ± 11.4	-
	Boiled (with skin)	56 ± 4	0.9 ± 0.0	72 ± 1.3	22.2 ± 3.8	70.5 ± 1.2
	Fried (with skin)	41 ± 3	0.7 ± 0.0	57 ± 1.1	43.8 ± 6.5	98.2 ± 5.8
Giant sea perch	Fresh (with skin)	54 ± 3	-	74 ± 3.0	37.1 ± 8.4	-
	Boiled (with skin)	65 ± 3	0.8 ± 0.2	73 ± 1.8	48.0 ± 21.2	100.0 ± 0.0
	Fried (with skin)	43 ± 3	0.7 ± 0.0	54 ± 2.0	52.9 ± 10.5	74.1 ± 1.7
Indo-Pacific Spanish mackerel	Fresh (with skin)	83 ± 5	-	75 ± 0.3	73.7 ± 8.7	-
	Boiled (with skin)	73 ± 0	0.6 ± 0.0	71 ± 2.1	72.6 ± 10.0	78.8 ± 0.0
	Fried (with skin)	60 ± 0	0.8 ± 0.0	58 ± 1.5	115.5 ± 5.1	100.0 ± 0.0
Short-bodied mackerel	Fresh (skinless)	52 ± 5	-	76 ± 1.8	108.8 ± 14.7	-
	Boiled (skinless)	47 ± 3	0.8 ± 0.0	71 ± 2.4	109.1 ± 28.4	88.8 ± 0.0
	Fried (skinless)	41 ± 3	0.8 ± 0.1	67 ± 4.9	76.9 ± 20.5	64.1 ± 0.0
Longtail tuna	Fresh (skinless)	66 ± 2	-	71 ± 2.2	198.5 ± 49.51	-
	Boiled (skinless)	58 ± 1	0.8 ± 0.0	68 ± 2.9	154.4 ± 44.5	64.4 ± 4.9
	Fried (skinless)	52 ± 3	0.8 ± 0.0	61 ± 0.6	262.4 ± 72.9	100.0 ± 0.0

**Table 4 foods-11-03312-t004:** Data on estimated marginal means of the interaction effects of several fish species and cooking methods on Se content and true retention of Se (*n* = 3).

Common Name	Se Content (μg/100 g of Product, Mean ± Standard Error)		True Retention of Se (%, Mean ± Standard Error)	
	Boiled	Fried	Boiled	Fried
Striped snakehead	22.2 ± 3.9 ^e,g^	43.8 ± 6.5 ^e,g,h^	70.7 ± 1.2 ^d^	96.8 ± 1.2 ^c,f^
Giant sea perch	48.0 ± 21.3 ^d,g^	52.9 ± 10.6 ^d,h^	100.0 ± 1.5 ^a,f^	74.1 ± 1.5 ^d^
Indo-Pacific Spanish mackerel	72.7 ± 10.1 ^c,f^	115.6 ± 5.1 ^b,f^	78.8 ± 1.5 ^c,f^	100.0 ± 1.5 ^a,f^
Longtail tuna	154.4 ± 44.6 ^a,f^	262.4 ± 72.9 ^a^	64.4 ± 1.5 ^e^	100.0 ± 1.5 ^b,f^
Short-bodied mackerel	109.2 ± 28.4 ^b,f^	76.9 ± 20.5 ^c,f,g^	88.8 ± 1.5 ^b,f^	64.1 ± 1.5 ^e^

For a particular variable, estimated marginal means values in the same column with different superscript letters were significantly different (*p* < 0.05 two-way ANOVA followed by Tukey’s HSD post hoc multiple comparisons).

**Table 5 foods-11-03312-t005:** Data on estimated marginal means of Se concentration and percentage of Se true retention by the major impacts of several kinds of fish and cooking methods (*n* = 3).

Common Name	Estimated Marginal Means ± Standard Error	
	Se (μg/100 g of Product)	True Retention (%)
Fish species:		
Striped snakehead	33.0 ± 12.7 ^c^	83.7 ± 4.4 ^a,b^
Giant sea perch	47.9 ± 14.7 ^c^	87.0 ± 5.4 ^a^
Indo-Pacific Spanish mackerel	89.7 ± 28.1 ^b^	89.4 ± 7.6 ^a^
Longtail tuna	208.4 ± 79.4 ^a^	82.2 ± 5.4 ^a,b^
Short-bodied mackerel	80.7 ± 3.2 ^b^	76.5 ± 7.6 ^b^
Cooking methods for several fish species:		
Fresh	89.5 ± 68.6 ^a,c^	-
Boiling	81.2 ± 51.9 ^b,c^	83.4 ± 19.5 ^b^
Frying	110.2 ± 89.4 ^a^	88.5 ± 18.2 ^a^

For a particular variable, values in the same column with various superscript letters of fish species or cooking techniques were significantly different (*p* < 0.05 two-way ANOVA followed by Tukey’s HSD post hoc multiple comparisons).

**Table 6 foods-11-03312-t006:** Estimated marginal means of % of in vitro bioaccessibility of Se by the major impacts of different fish species and cooking techniques (*n* = 3).

Common Name	Estimated Marginal Means ± Standard Error	
	Boiling (%)	Frying (%)
Fish species:		
Striped snakehead	70.0 ± 0.6 ^a^	58.5 ± 2.8 ^a^
Giant sea perch	57.7 ± 3.8 ^a,b^	48.8 ± 7.2 ^a^
Indo-Pacific Spanish mackerel	64.6 ± 3.9 ^a^	51.8 ± 1.3 ^a^
Longtail tuna	49.0 ± 1.1 ^b^	49.1 ± 4.0 ^a^
Short-bodied mackerel	52.8 ± 3.9 ^a,b^	47.1 ± 1.4 ^a^
Cooking methods for several fish species:		
Average	58.8 ± 8.5 ^a^	51.1 ± 4.4 ^a^

For a given variable, values in the same column with differing superscript letters of fish species or cooking methods in the same row were significantly different (*p* < 0.05 two-way ANOVA followed by Tukey’s HSD post hoc multiple comparisons).

## Data Availability

The data are included within the article.

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
