# Peer review of "In Vitro Bioaccessibility of Selenium from Commonly Consumed Fish in Thailand"

_foods, 2022, doi:10.3390/foods11213312_

Round 1

Reviewer 1 Report

Authors have selected and worked on a very interesting topic and summarize it very well. But there are few suggestions that should must be addressed.

Please rephrase the first line of abstract as grammatically it is not correct… Selenium (Se) is a crucial trace element for humans which fish are showed high Se content.

Authors should must mention about the short treatment plan in abstract.

Authors should also mention about the statistical analysis in abstract.

Please rephrase this sentence … No significant difference in bioaccessibility was found among fried fish and between cooking methods . it can be like (No significant difference in bioaccessibility was found among methods of preparation i.e. cooking and frying)

In line 44 introduction section. Please mention those studies who have revealed about the high content of Se in Fish or you can give references of those studies.

Methodology is well explained.

Results are little bit confusing, please explain the graphs and  tables in results explanation.

Give proper justification and comparison in discussion section.

Plagiarism is 27 % authors are encouraged to please remove it less than 18 %.

Please recheck your references according to journal guidelines and format.

Authors are requested to please thoroughly read the article and improve its grammatical mistakes.

Author Response

Comments and Suggestions for Authors

Response to Comments and Suggestions

Please rephrase the first line of abstract as grammatically it is not correct… Selenium (Se) is a crucial trace element for humans which fish are showed high Se content.

The sentence has been revised.

Authors should must mention about the short treatment plan in abstract.

The short plan has been mentioned in abstract.

Authors should also mention about the statistical analysis in abstract.

The statistical analysis has been mentioned in the abstract (two-way ANOVA with interaction followed by Turkey’s HSD post hoc test).

Please rephrase this sentence … No significant difference in bioaccessibility was found among fried fish and between cooking methods . it can be like (No significant difference in bioaccessibility was found among methods of preparation i.e. cooking and frying)

This sentence has been amended.

In line 44 introduction section. Please mention those studies who have revealed about the high content of Se in Fish or you can give references of those studies.

References indicating the high Se content in fish are given (references 9-12)

Methodology is well explained.

-

Results are little bit confusing, please explain the graphs and tables in results explanation.

The authors have added explanations to graphs and tables.

Give proper justification and comparison in discussion section.

-

Plagiarism is 27 % authors are encouraged to please remove it less than 18 %.

The manuscript has been revised and the plagiarism rate has reached the journal’s criteria.  Most of the plagiarism are come from the part of material and method which are technical term of analysis or fish name, for instance, Inductively Coupled Plasma Mass Spectrometry (ICP-QQQ-MS), Indo-pacific Spanish mackerel, Longtail tuna, and Short-bodied mackerel, etc. Could you please consider and accept tham.

Please recheck your references according to journal guidelines and format.

The format of references has been rechecked.

Authors are requested to please thoroughly read the article and improve its grammatical mistakes.

The grammatical mistakes in this paper have been corrected by a native English speaker/editor from the USA.

Reviewer 2 Report

The study aims to investigate the bioaccessibility of Se based on the in vitro equilibrium method for commonly consumed fish in Thailand. 

The Introduction presents briefly informations about Se and fishes as well as mentioning some considerations about bioaccessibility. Studies on Se bioaccessibility and methods to simulate in vitro digestion have been poorly explorated. The term "  in vitro equilibrium method" for bioaccessibility assays are used but the authors did not cite any references about this method used. "Equilibrium method" would be dialysis method?

Materails and methods

Why the samples were freeze-dried? I believe the fish will not be consumed freeze-dried and this fact can be affect the bioaccessibility results?

Why the weights of all fish before and after removing inedible parts were recorded? In addition, why Weight yield factor was estimated? This step is very confuse along the text. 

For in vitro digestion assay the authors cited Miller (1981). Many adaptations to Miller's proposed method for dialysis trials have already been presented in publications that are more recent. Why did you choose this reference? We are in 2022.

Results and discussion:

The results are presented in confusing ways. What means estimated marginal means?

The discussion presented in topic 4.3  In vitro bioaccessibility of Se in fish is very poor. Little was actually discussed about selenium, comments about effect of major components such as proteins, fiber, carbohydrates and lipids in the samples as data not presented. Confusing comments also appear about other elements, mercury. 

The paper needs major revision.

  •  
  •  

Author Response

Comments and Suggestions for Authors

Response to Comments and Suggestions

The Introduction presents briefly informations about Se and fishes as well as mentioning some considerations about bioaccessibility. Studies on Se bioaccessibility and methods to simulate in vitro digestion have been poorly explorated. The term "  in vitro equilibrium method" for bioaccessibility assays are used but the authors did not cite any references about this method used. "Equilibrium method" would be dialysis method?

The authors provided concise information for facts that are well known, while also covering all necessary background data.

The term in vitro bioaccessibility has been cited in reference 14. 

The authors cited examples of studies that have used the dialysis method in references 15-17.

The word "Equilibrium method" has been revised.

Why the samples were freeze-dried? I believe the fish will not be consumed freeze-dried and this fact can be affect the bioaccessibility results?

For analytical purposes, only a small portion (about 1 g) is used for analysis as representative of each sample. Samples must be prepared as fine particles before weighing using the freeze-dried system. After analysis, all samples are calculated back into a fresh sample or product using the moisture loss during the freeze drying system. Consequently, bioaccessibility is not affected in the freeze-dried samples.

Why the weights of all fish before and after removing inedible parts were recorded? In addition, why Weight yield factor was estimated? This step is very confuse along the text. 

The weights before and after removing the inedible parts must be recorded because they are used for calculating the % edible portion for each fish. For yield factor, this recorded data are used to calculate the true retention of Se (Se content before and after cooking).

For in vitro digestion assay the authors cited Miller (1981). Many adaptations to Miller's proposed method for dialysis trials have already been presented in publications that are more recent. Why did you choose this reference? We are in 2022.

The authors have amended the citation to reflect recent studies.

Results and discussion:

The results are presented in confusing ways. What means estimated marginal means?

A marginal mean is the mean response for each category of a factor, adjusted for any other variables in the model. It has the advantage of identifying the means and their standard errors to interpret results. The difference in those means is what measures the effect of the factor. While that difference can also appear in the regression coefficients, looking at the means themselves gives the data a context and makes interpretation more straightforward.

The discussion presented in topic 4.3  In vitro bioaccessibility of Se in fish is very poor. Little was actually discussed about selenium, comments about effect of major components such as proteins, fiber, carbohydrates and lipids in the samples as data not presented. Confusing comments also appear about other elements, mercury. 

The discussion of this topic has been revised.

Reviewer 3 Report

The manuscript reports studies on the effect of cooking and frying of five different species of fish commonly consumed in Thailand (Striped snakehead, Giant sea perch, Indo-pacific Spanish mackerel, Longtail tuna, and Short-bodied mackerel) with regard to total and bioaccessible Se content. The study is very relevant in exploring alternative nutritional sources and evaluating different methods of processing this food. The methodology is adequate and sufficiently developed, however, I suggest some minor improvements highlighted below:

1. INTRODUCTION:

Line 67. The word “volume” should be replaced by “content”.

Line 97. The sentence should be rewritten, as the ICP-MS has lower detection limit values instead of “superior detection limit” as it is written.

2. MATERIALS AND METHODS

Line 143. Please detail the grinding procedure.

Line 168. Add the unit of HNO3 percentage.

Line 175. Add the unit to the value “0.25”

Line 176. Add the unit of HNO3 percentage. Is this w/v? Please review the entire text.

Line 267. Replace “precision” with “accuracy”

Line 268. Replace “optimum Se concentration” with “experimental Se concentration”

3. RESULTS

Line 336. Figure 1. The Y-axis label must be added in order to idenfify which variable the y-axis represents and its unit.

Line 362 - 365. The sentence “The data on %TR represented the percentage loss of Se in cooked fish as compared to the Se content before cooking” should be rewritten as it does not meet the definition described in Line 196 of the manuscript, in which the TR is described as “the proportion of a nutrient that remains in a cooked foodstuff as compared to the original quantity prior to cooking.”

> ADDITIONAL COMMENTS <

- Add to the manuscript the brand and purity of the O2 and He used as reaction and collision gases.

- Please replace the term “Heavy Metals” with “Potentially Toxic Elements” throughout the text.

- Details of the freeze dryer and shaking water bath, such as brand and model, must be added to the text.

Author Response

Comments and Suggestions for Authors

Response to Comments and Suggestions

1. INTRODUCTION:

Line 67. The word “volume” should be replaced by “content”.

Manuscript was modified according to the reviewer’s comments.

Line 97. The sentence should be rewritten, as the ICP-MS has lower detection limit values instead of the “superior detection limit” as it is written.

Manuscript was modified according to the reviewer’s comments.

2. MATERIALS AND METHODS

Line 143. Please detail the grinding procedure.

The detail of grinding was added according to the reviewer’s comment.

Line 168. Add the unit of HNO3 percentage.

The unit was added (v/v) according to the reviewer’s comment.

Line 175. Add the unit to the value “0.25”

The unit was added (g) according to the reviewer’s comment.

Line 176. Add the unit of HNO3 percentage. Is this w/v? Please review the entire text.

The unit added the entire text (v/v) according to the reviewer’s comment.

Line 267. Replace “precision” with “accuracy”

The manuscript was modified according to the reviewer’s comments.

Line 268. Replace “optimum Se concentration” with “experimental Se concentration”

Manuscript was modified according to the reviewer’s comments.

3. RESULTS

Line 336. Figure 1. The Y-axis label must be added in order to identify which variable the y-axis represents and its unit.

The Y-axis labels are added to the Figure description (both A and B).

Line 362 - 365. The sentence “The data on %TR represented the percentage loss of Se in cooked fish as compared to the Se content before cooking” should be rewritten as it does not meet the definition described in Line 196 of the manuscript, in which the TR is described as “the proportion of a nutrient that remains in a cooked foodstuff as compared to the original quantity prior to cooking.”

Line 362-365 is revised as “The data on %TR represented the percentage of Se that remained in cooked fish as compared to the Se content before cooking”

> ADDITIONAL COMMENTS <

- Add to the manuscript the brand and purity of the Oand He used as reaction and collision gases.

Manuscript was modified according to the reviewer’s comments.

- Please replace the term “Heavy Metals” with “Potentially Toxic Elements” throughout the text.

Manuscript was modified according to the reviewer’s comment.

- Details of the freeze dryer and shaking water bath, such as brand and model, must be added to the text.

Manuscript was modified according to the reviewer’s comments.
